# Recent Status and Methodological Quality of Return-to-Work Rates of Cancer Patients Reported in Japan: A Systematic Review

**DOI:** 10.3390/ijerph16081461

**Published:** 2019-04-24

**Authors:** Atsuhiko Ota, Akiko Fujisawa, Kenji Kawada, Hiroshi Yatsuya

**Affiliations:** 1Department of Public Health, Fujita Health University School of Medicine, 1-98 Dengakugakubo, Kutsukake-cho, Toyoake, Aichi 470-1192, Japan; akifuji@fujita-hu.ac.jp (A.F.); yatsuya@fujita-hu.ac.jp (H.Y.); 2Department of Medical Oncology, Fujita Health University School of Medicine, 1-98 Dengakugakubo, Kutsukake-cho, Toyoake, Aichi 470-1192, Japan; kekawada@fujita-hu.ac.jp

**Keywords:** cancer, methodological quality assessment, Japan, return-to-work rate, systematic review

## Abstract

Cancer patients’ return-to-work rates in Japan and their methodological quality have been little reported. We conducted a systematic review to explore the recent return-to-work rates and to assess the methodological quality of the existing literature. We selected 13 papers (2 in English and 11 in Japanese) published between 2005 and 2017. The return-to-work rates ranged from 53.8% to 95.2%. Of the selected papers, 12 papers employed a cross-sectional design, possessing high risk of selection bias due to participant selection. A total of 8 papers did not fully report the subjects’ sex, age, employment status at cancer diagnosis, cancer site, stage, and treatment, suggesting high risk of selection bias due to confounding variables. High or unclear risk of attrition bias due to incomplete outcome data was detected in 12 papers in which data on return to work were not collected from all participants. High risk of reporting bias due to selective outcome reporting was pointed out in 6 studies in which the subjects’ employment status at return to work or the duration between cancer diagnosis and assessment of return to work was unclear. Future studies must reduce the risk of selection, attrition, and reporting biases for specifying accurate return-to-work rates.

## 1. Introduction

In Japan, an estimated one million people develop cancer annually [1,2]. About 30% of the incidence occurs in the working-age population, ages 15–64 [2]. Although cancer remains the leading cause of death in Japan, its age-adjusted mortality rate is decreasing [1]. Working-age people are highly likely to survive cancer. Among cancer patients diagnosed between 2006 and 2008, five-year relative survival rates were estimated as 79.5%, 72.6%, and 65.5% for those aged 15–44, 45–54, and 55–64, respectively [3]. Given this, it is necessary to help working-age cancer patients return to work for their social participation. The Japanese government addresses this issue in the latest Basic Plan to Promote Cancer Control Programs based on the Cancer Control Act [1]. A systematic review reported that cancer patients were 1.37 times more likely to be unemployed than the general population and non-cancer patients [4].

Return-to-work rates indicate how many working-age cancer patients succeed in returning to work. Systematic reviews have elucidated these rates [4,5,6]. Mehnert reported that 63.5% (range: 24–94%) of cancer patients returned to work [5]. Paltrinieri et al. reported the rates ranging from 39% to 77% [6]. No Japanese evidence was employed in the previous reviews. This evoked questions of whether recent return-to-work rates of cancer patients in Japan were unclear and whether the quality of existing literature on the rates was too low to merit inclusion. Research papers are the only source to infer the return-to-work rates of cancer patients in Japan since there is no governmental surveillance system to determine them. In this systematic review, we explored the recent return-to-work rates of cancer patients in Japan and assessed the methodological quality of existing relevant literature.

## 2. Methods

We referred to the PRISMA (Preferred Reporting Items for Systematic Reviews and Meta-Analyses) Statement [7].

### 2.1. Literature Search

We searched the existing literature using the PubMed, Scopus, CINAHL, and ICHUSHI web databases. ICHUSHI contains biomedical journals and publications published in Japan. The synthesis of searching terms was “cancer” (*gan* in Japanese) AND (“return to work” (*fukusyoku*) OR “employment” (*shurou*)) AND “Japan.” The term “Japan” was omitted for the ICHUSHI search. Inclusion criteria for the present review were as follows: (1) Studies were carried out in Japan; (2) Cancer diagnosis could be confirmed by a patient chart or any other objective records, such as a doctor’s note; (3) Subjects were working at the time of diagnosis; (4) Return-to-work rates were presented; (5) The paper was written in English or Japanese; and (6) The paper was published from 2005 to 2017, considering the progress of cancer treatment. For criterion (3), we assumed that cancer diagnosis was confirmed objectively by a patient chart in hospital-based studies. Hospital-based studies collected the eligible participants at hospitals, whereas worksite-based studies did it at worksites. We additionally identified eligible papers by referring to the extracted papers. The literature search was independently conducted by the authors A.O. and A.F. Disagreement was resolved by consensus.

### 2.2. Return-to-Work Rates

We re-calculated the return-to-work rate for each selected study, limiting ourselves to data about the subjects who were working at the time of cancer diagnosis. Some of the selected studies had small sample sizes. Point return-to-work rates estimated in such studies might not reflect the rate of the corresponding population. Therefore, we calculated a 95% confidence interval (CI) for each study’s return-to-work rate. We used the following formula, where $\hat{p}$ is the return-to-work rate and *n* is the corresponding number of subjects in each study:(1)$$\hat{p}-1.96\sqrt{\frac{\hat{p}\left(1-\hat{p}\right)}{n}}<\;95\%\;CI\;of\;\hat{p}<\;\hat{p}+1.96\sqrt{\frac{\hat{p}\left(1-\hat{p}\right)}{n}}$$

We calculated the rates not only for the total of all cancers but also for gastric, intestinal (small intestine, colon, and rectum), female genital (uterus and ovary), and breast cancer since three or more selected studies addressed these kinds of cancer.

### 2.3. Methodological Quality Assessment

Potential selection, performance, detection, attrition, and reporting biases in each selected study were evaluated using the Risk of Bias Assessment Tool for Non-randomized Studies (RoBANS) [8] since this systematic review did not include non-randomized studies or intervention studies. The RoBANS assesses six domains. Domains 1, “Selection of participants (selection bias),” and 2, “Confounding variables (selection bias),” evaluate possibilities of inadequate selection of participants and consideration of confounding variables. For Domain 1, risk of selection bias due to selection of participants was evaluated based on the study design: risk was low for prospective studies and high for cross-sectional studies. Prospective designs determined the participants at the time of cancer diagnosis and followed them consecutively. Cross-sectional studies determined them and collected all data upon examining their return to work. For Domain 2, we defined sex, age, employment status at cancer diagnosis (type of occupation, full-time/part-time), cancer site, stage, and treatment (operation, chemotherapy, radiotherapy) as the confounding variables. We referred to the existing literature to determine the confounding variables [3,9,10]. Studies that reported all confounding variables had a low risk of selection bias due to disclosure of confounding variables. Domain 3, “Measurement of exposure (performance bias),” evaluates whether measurement of exposure (i.e., cancer diagnosis) is objectively carried out. Studies in which cancer diagnosis was confirmed by medical or other objective records had a low risk of the bias. Domain 4, “Blinding of outcome assessment (detection bias),” evaluates how to assess the outcome, namely, the return to work. Risk of the bias was high when researchers subjectively assessed subjects’ return to work, and low when return to work was assessed using objective records or participants’ self-reports. The risk was unclear for studies which did not describe the way of determining subjects’ return to work. For Domain 5, “Incomplete outcome data (attrition bias),” risk of the bias was low in studies that collected data on return to work from all participants and high in studies that did not. For the studies at high risk, we calculated the missing response rate, that is, for non-respondents regarding outcome (return-to-work) assessment in the eligible participants. The eligible participants were not limited to workers in some studies. For such studies, we calculated the missing response rate, including those who were not working at the time of cancer diagnosis. The bias risk was unclear in the case where the number of the eligible participants was not specified and only that of the subjects analyzed was presented. For Domain 6, “Selective outcome reporting (reporting bias),” risk of the bias was low in studies that fully reported the return-to-work rates, employment status at the time of return to work, and duration between cancer diagnosis and assessment of return to work. This series of methodological quality assessments was executed by the author A.O. and checked by A.F. Disagreement was resolved by consensus.

## 3. Results

### 3.1. Literature Search

We extracted 510 papers through database searching (Figure 1). Duplicate extraction occurred for 108 papers. We excluded 390 papers for the following reasons: case report, review, or conference abstract; study conducted outside Japan; laboratory experimentation; subjects not of working age; return-to-work rates not reported; and data duplicated with other studies. We added one more paper, referring to the extracted papers. Eventually, 13 papers were selected for review (Table 1) [11,12,13,14,15,16,17,18,19,20,21,22,23]. Only two papers were written in English [16,19]; the remaining 11 were written in Japanese [11,12,13,14,15,17,18,20,21,22,23], although English abstracts were available for some [13,14,15,17,22,23].

### 3.2. Return-to-Work Rates

Return-to-work rates ranged from 53.8% to 95.2% for the total of all cancers (Figure 2), from 42.9% to 93.3% for gastric cancer, from 66.7% to 84.2% for intestinal cancer, from 42.9% to 95.2% for female genital cancer, and from 45.0% to 89.7% for breast cancer (Figure 3).

### 3.3. Methodological Quality Assessment

#### 3.3.1. RoBANS Domain 1 “Selection of Participants (Selection Bias)”

Only one study by Endo et al. [19] explicitly indicated that employees with an episode of absence due to sickness were determined as participants at the time of cancer diagnosis and were followed (Table 1 and Table 2). We regarded this study’s design as prospective and worksite-based, and consequently, its bias risk was low. All the other studies were cross-sectional and hospital-based [11,12,13,14,15,16,17,18,20,21,22,23]. They determined the participants at the time of examination of return to work. Their bias risk was judged as high.

#### 3.3.2. RoBANS Domain 2 “Confounding Variables (Selection Bias)”

Some studies did not report subjects’ employment status at diagnosis [11,12,13,14,15], cancer site [16], stage [11,12,13,15,19,20], or treatment [19]. They were regarded as having a high bias risk. The other studies fully reported these variables [17,18,21,22,23]. They were at low bias risk.

#### 3.3.3. RoBANS Domain 3 “Measurement of Exposure (Performance Bias)”

The bias risk was low for all selected studies. A worksite-based study by Endo et al. ascertained cancer diagnosis based on sickness insurance system claims accompanied by a doctor’s note [19]. The other studies were hospital-based and assumed that cancer diagnosis was objectively confirmed [11,12,13,14,15,16,17,18,20,21,22,23].

#### 3.3.4. RoBANS Domain 4 “Blinding of Outcome Measurement (Detection Bias)”

The bias risk was unclear in a study by Shimada et al. [12]. They did not explicitly describe the way of determining subjects’ return to work. The other studies were at low bias risk. Return to work was confirmed by sickness insurance system records [19] or subjects’ self-reports [11,13,14,15,16,17,18,20,21,22,23].

#### 3.3.5. RoBANS Domain 5 “Incomplete Outcome Data (Attrition Bias)”

A prospective study by Endo et al. was at low bias risk as they succeeded in collecting outcome data of all participants [19]. In some studies, data on return to work were not collected from all participants [11,13,14,16,18,20,21,22,23]. These studies were evaluated as at high bias risk. The missing response rates ranged from 5.7 to 70.3% (median: 32.4%). In some studies, the number of eligible participants was unavailable and only the number of the subjects analyzed was reported [12,15,17]. The bias risk was unclear for these studies.

#### 3.3.6. RoBANS Domain 6 “Selective Outcome Reporting (Reporting Bias)”

Some studies fully reported return-to-work rates, employment status at return to work, and duration between cancer diagnosis and assessment of return to work and were at low bias risk [11,16,18,19,21,22,23]. The others were at high bias risk because of incomplete disclosure of employment status at return to work [13,14,15] or duration between diagnosis and return-to-work assessment [12,17,20].

## 4. Discussion

The return-to-work rates of cancer patients reported in Japan varied from 53.8% to 95.2% for the total of all cancers. The variation existed even when the subjects were classified by cancer site (gastric, intestinal, female genital, and breast cancer). Of course, this can partly be explained by the differences in the subjects’ characteristics among the studies. At the same time, we found high risk of selection bias due to participant selection and confounding variables, attrition bias due to incomplete outcome data, and reporting bias due to selective outcome reporting in the selected studies. We discuss below the possible effects of these biases on the reports of return-to-work rates.

We regarded almost all the selected studies as having a high risk of selection bias due to selection of participants. Except for one prospective worksite-based study [19], all the selected studies employed a cross-sectional design, determining subjects at the time of examination of return to work [11,12,13,14,15,16,17,18,20,21,22,23]. In these cross-sectional studies, cancer patients who had died by the time of recruitment were not included in the calculation of return-to-work rates. Then, the rates were overestimated. This is the fatal flaw in specifying return-to-work rates.

We regarded the studies which did not fully disclose the subjects’ employment status at diagnosis [11,12,13,14,15], cancer site [16], stage [11,12,13,15,19,20], or treatment [19] as having a high risk of selection bias due to confounding variables. Since cancer patients’ characteristics at cancer diagnosis affect return-to-work rates, concealing them evokes a skepticism of overestimating return-to-work rates. Some studies only focused on patients with special characteristics, which must be considered in interpreting the rates. For example, Uchida et al. only examined breast cancer patients who took axillary lymph node dissection (ALND) and showed a comparatively low return-to-work rate, 56.3% [13]. A recent French prospective study specified ALND as a risk factor of a longer duration of sick leave [24]. Endo et al. presented a high return-to-work rate of breast cancer patients, 89.7% [19]. This could be because the study’s subjects were employees of large-scale companies with occupational health professionals. Roelen et al. showed that employees of large companies returned to work earlier than those of small companies; they assumed that large companies were better able to accommodate working conditions [25]. Previous research notes that occupational health support is helpful in returning to work [26,27]. Occupation type also affects the return-to-work rates and whether they return to their former jobs. Matsuda et al. [18] and Shionoya et al. [20] reported that civil servants or full-time employees were more likely to successfully return to their former job. Cancer site, stage, or treatment are essential factors in interpreting cancer patients’ return-to-work rates since they determine survivability from cancer [3].

Regarding attrition bias due to incomplete collection of data, responses on return to work were missing in many of the selected studies [11,13,14,18,20,21,22,23]. The missing response rates ranged from 5.7 to 70.3% with the median being 32.4%. A drawback is that return-to-work rates were overestimated in such studies by excluding the non-respondents from the calculation. This overestimation became worse if those who did not return to work were more unlikely to respond than those who did return.

Regarding reporting bias due to selective outcome reporting, some studies did not report the employment status at return to work [13,14,15]. An absence of reporting employment status at return to work is an obstacle when specifying the definition of return to work and meta-analyzing the return-to-work rates. The return-to-work rates increased when being a homemaker was regarded as a successful return to work [14,18]. Nakamura et al. noted a significant decrease in wages for cancer patients who returned to employment other than their former job [21]. This economic disadvantage could discourage some cancer patients from returning to work if their former job were unavailable. Most of the selected studies disclosed the duration between cancer diagnosis and assessment of return to work [11,13,14,15,16,18,19,21,22,23]. However, we have to point out that, in the cross-sectional studies, the duration between cancer diagnosis and return-to-work assessment varied greatly, ranging from one month to longer than 10 years, both between studies and even between subjects in a single study [11,13,14,16,17,18,21,22,23]. This makes the interpretation of the return-to-work rates puzzling. If the duration was long, cancer survivors became old. Their return-to-work rates could be skewed because of health disorders (which were not cancer comorbidity) or the Japanese custom of compulsory retirement based on age. If the duration was short, cancer patients might not recover sufficiently, resulting in a low return-to-work rate. Suzuki and Itou evaluated return to work of the patients only one month after the operation and reported a low return-to-work rate, 42.9%, of patients with gastric cancer [15]. It is difficult to standardize the duration between cancer diagnosis and return-to-work assessment in a cross-sectional study which determines the participants upon examining their return to work. Cross-sectional design is thus inappropriate for the assessment of the return-to-work rates. Of our 13 selected studies, 11 were cross-sectional studies. Therefore, we abandoned meta-analyzing the return-to-work rates in the present study.

Some may say that the ROBINS-I (Risk Of Bias In Non-randomized Studies—of Interventions) [28] should have been used instead of the RoBANS to assess the risk of biases. We do not believe that the use of the RoBANS is a disadvantage in assessing the risk of biases for the present study. The ROBINS-I was originally developed to assess the risk of biases in non-randomized studies examining the effects of interventions. The 13 selected studies for our study consisted of 12 cross-sectional studies and one prospective study. Both the RoBANS and ROBINS-I shared a risk assessment of selection, performance, detection, attrition, and reporting biases. Using the RoBANS, we could point out a high risk of selection, attrition, and reporting biases existing in the 13 selected studies.

What future studies must do is to improve the study quality by reducing the risk of selection, attrition, and reporting biases in order to correctly find the return-to-work rates. Regarding selection bias due to participant selection, a cross-sectional design should be avoided, although it is also true in other countries that a cross-sectional design has often been employed in previous studies to estimate return-to-work rates of cancer patients [4,5,6]. Instead, prospective cohort studies must be conducted to determine all eligible participants at the time of cancer diagnosis. Some studies employed a prospective cohort design to examine the return to work of patients with injuries even in the 1980s and 1990s [29] and that of cancer patients recently [30,31,32,33,34]. Disclosure of the subjects’ characteristics in detail is necessary to eliminate potential selection bias due to confounding variables and reporting bias due to selective outcome reporting. Adequate trial registration and adherence to reporting guidelines would limit selective reporting of the confounding variables and outcome [35]. Researchers must register their trials prospectively, define the confounding variables and study outcomes explicitly, and address discrepancies between their own findings and existing ones honestly [35]. To avoid attrition bias, researchers must pay greater attention to collecting data on the outcome (return to work) from all participants. A potential solution is a collaboration with the national database which involves all citizens. For example, researchers in Korea examined the employment status of cancer patients using the National Health Insurance administrative database [30,36,37]. This contributed to reducing the risk of attrition bias. No Japanese relevant studies have used such official data. In addition, future studies must report in English for enabling comparison of cancer patients’ return-to-work rates with other countries. Only two of the 13 selected studies were written in English [16,19]. This suggests a weak international influence of existing Japanese evidence. This must be why Japanese evidence was little employed in previous systematic reviews regarding cancer patients’ return-to-work rates [4,5,6]. We used major and reliable web databases to extract existing papers. However, we could possibly have failed to detect a few relevant papers which were written in English and published in little-known journals.

## 5. Conclusions

We conducted a systematic review to explore the recent return-to-work rates of cancer patients in Japan and to assess the methodological quality of the existing literature. The return-to-work rates reported in the 13 selected papers ranged from 53.8% to 95.2%. We found high risk of selection bias due to participant selection and confounding variables, attrition bias due to incomplete outcome data, and reporting bias due to selective outcome reporting, which were an obstacle in correctly identifying the return-to-work rates. Future studies must be planned to eliminate the risk of these biases.

## Figures and Tables

**Figure 1 ijerph-16-01461-f001:**
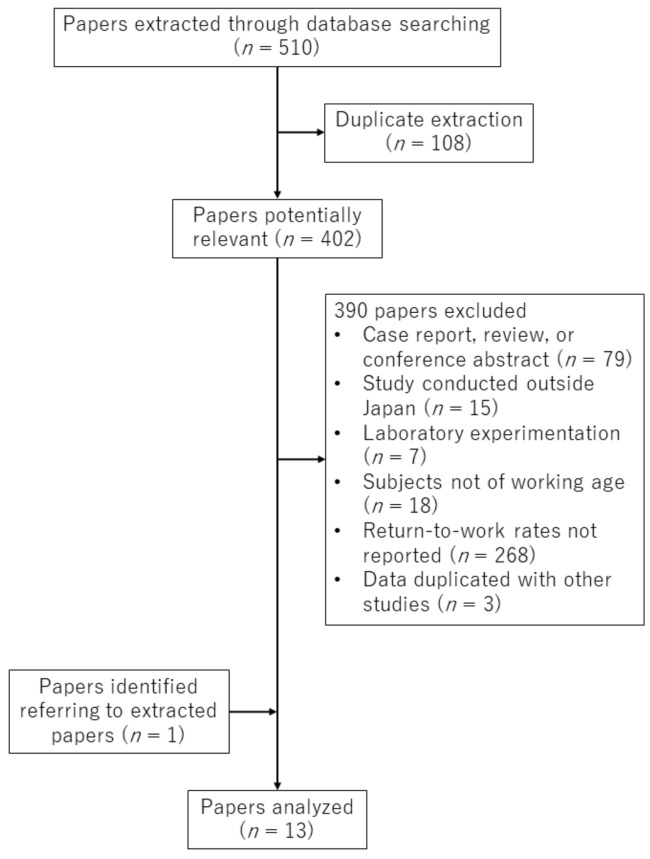
Paper selection process.

**Figure 2 ijerph-16-01461-f002:**
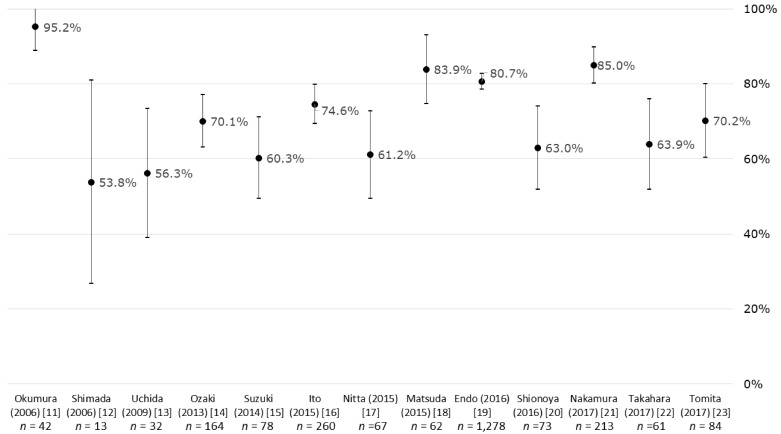
Return-to-work rates and 95% confidence intervals for cancer patients in Japan.

**Figure 3 ijerph-16-01461-f003:**
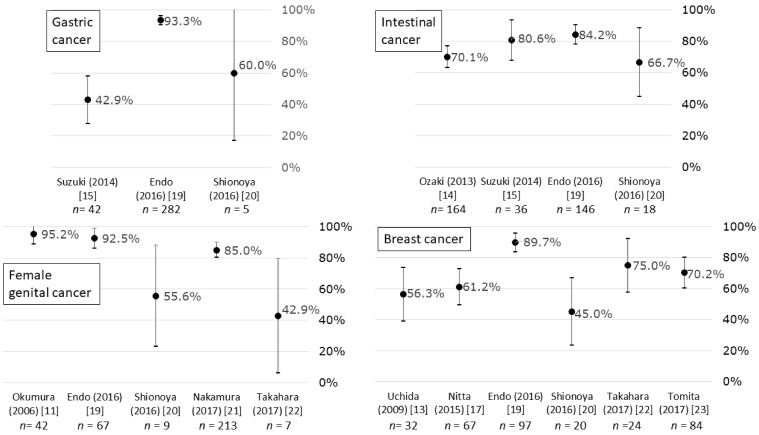
Return-to-work rates and 95% confidence intervals for (upper left) gastric, (upper right) intestinal (small intestine, colon, and rectum), (lower left) female genital (uterus and ovary), and (lower right) breast cancer patients in Japan.

**Table 1 ijerph-16-01461-t001:** Summary of selected studies.

First Author (Publication Year)	Study Design and Setting	At Cancer Diagnosis		(1) Cancer Site, (2) Stage, (3) Treatment	At Return to Work
		(1) Number, (2) Sex, (3) Age	Employment Status		(1) Return-to-Work Rate, (2) Employment Status among Those Who Returned to Work, (3) Duration between Cancer Diagnosis and Assessment of Return to Work
Okumura (2006) [11]	C, H	(1) 42, (2) Female only, (3) ^‡^ Mean (SD): 53.8 (23.5), range: 27–74	NA	(1) Female genitals, (2) NA, (3) ^‡^ Ope: 58%, Ope+Che and/or Rad: 42%	(1) 95.2%, (2) Working style was reported. Standing work: 60%, sedentary work: 33%, (3) Return to work <1 month after Ope: 25%, <3 months: 35%, <6 months: 20%
Shimada (2006) [12]	C, H	(1) 13, (2) ^‡^ Male: 61%, (3) ^‡^ Mean 58, range: 39–73	NA	(1) Head/neck, (2) NA, (3) Ope (RND): 100%	(1) 53.8%, (2) Former work, (3) NA
Uchida (2009) [13]	C, H	(1) 32, (2) Female only, (3) ^‡^ Mean: 52, range: 28–73	NA	(1) Breast, (2) NA, (3) Ope (ALND): 100%, Che: 0%	(1) 56.3%, (2) NA, (3) Mean: 23 months, range 4–38
Ozaki (2013) [14]	C, H	(1) 164, (2) Male: 65%, (3) ^‡^ 50–59: 31%, 60–69: 53%	NA	(1) Colon, (2) ^‡^ I: 14%, II: 28%, III: 30%, IV: 20%, NA: 8%, (3) ^‡^ LT: 78%, LS: 17%	(1) Total: 70.1%, ^†^ 51.8%, (2) All occupations including homemaker, (3) Range: 100–2000 days
Suzuki (2014) [15]	C, H	(1) 78, (2) ^‡^ Male: 73%/51% (stomach/colon), (3) Mean: 56.2/57.0 (stomach/colon)	NA	(1) 54%/46% (stomach/colon), (2) NA, (3) LT: 43%/33%, LS: 57%/67% (stomach/colon)	(1) 42.9%/80.6% (stomach/colon), (2) NA, (3) 1 month after Ope
Ito (2015) [16]	C, H	(1) 260, (2) Male: 50%, (3) Mean (SD): 54.9 (8.3)	Regularly employed: 48%, self-employed: 25%, non-regularly employed: 15%	(1) NA, (2) Early: 34%, advanced: 54%, NA: 12%, (3) Ope: 72%, Che: 63%, Rad: 36%	(1) Total: 74.6%, (2) Regularly employed: 47%, self-employed: 27%, non-regularly employed: 15%, (3) Mean (SD): 4.2 (3.5) years
Nitta (2015) [17] (new onset, Ope+Che)	C, H	(1) 17, (2) Female only, (3) ^‡^ Median: 54, range: 31–75	Regularly employed: 41%, part-time: 29%, self-employed: 24%	(1) Breast, (2) ^‡^ I: 17%, II: 70%, III: 13%, (3) ^‡^ Ope+Che (including MTT): 100%	(1) 41.2%, (2) Former work: 86%, (3) NA
Nitta (2015) [17] (new onset, Ope+ET)	C, H	(1) 17, (2) Female only, (3) ^‡^ Median: 55, range: 34–74	Regularly employed: 35%, part-time: 41%, self-employed: 18%	(1) Breast, (2) ^‡^ I: 26%, II: 59%, III: 7%, (3) ^‡^ Ope+ET: 100%	(1) 64.7%, (2) Former work: 100%, (3) NA
Nitta (2015) [17] (recurrence, Che)	C, H	(1) 11, (2) Female only, (3) ^‡^ Median: 51, range: 33–67	Regularly employed: 27%, part-time: 45%, self-employed: 18%	(1) Breast, (2) ^‡^ I: 10%, II: 45%, III: 20%, NA: 25%, (3) ^‡^ Ope: 95%, Che (including MTT): 100%	(1) 45.5%, (2) Former work: 60%, (3) Median: 51 months, range: 1–132 months
Nitta (2015) [17] (recurrence, ET)	C, H	(1) 10, (2) Female only, (3) ^‡^ Median: 55, range: 36–78	Regularly employed: 20%, part-time: 50%, self-employed: 20%	(1) Breast, (2) ^‡^ I: 27%, II: 36%, III: 23%, NA: 14%, (3) ^‡^ Ope: 82%, ET: 100%	(1) 70.0%, (2) Former work: 71%, (3) Median: 97 months, range: 3–180 months
Nitta (2015) [17] (follow-up)	C, H	(1) 12, (2) Female only, (3) ^‡^ Median: 51, range: 33–67	Regularly employed: 50%, part-time: 17%, temporary: 25%	(1) Breast, (2) ^‡^ I: 55%, II: 27%, III: 14%, (3) ^‡^ Ope: 100%, Che: 50%, ET: 50%, Rad: 73%	(1) 91.7%, (2) Former work: 73%, (3) Median: 72 months, range: 5–120 months
Matsuda (2015) [18]	C, H	(1) 62, (2) ^‡^ Male: 42%, (3) ^‡^ 30–39: 11%, 40–49: 17%, 50–59: 32%, 60–69: 30%	Regularly employed: 39%, self-employed: 23%, homemaker: 19%, part-time: 10%, civil servant: 8%	(1) ^‡^ Breast: 24%, lung: 20%, stomach: 11%, uterine: 9%, colon: 5%, other: 32%, (2) ^‡^ I: 24%, II: 9%, III: 14%, IV: 20%, NA: 33%, (3) ^‡^ Ope: 58%, Che: 83%, Rad: 23%	(1) Total: 83.9%, ^†^ 59.7%, (2) Regularly employed: 33%, self-employed: 17%, homemaker: 29%, part-time: 6%, civil servant: 10%, (3) ^‡^ <2 years: 65%, 2–6 years: 20%, 7–9 years: 11%, 10+ years: 3%
Endo (2016) [19]	P, W	(1) 1278, (2) Male: 81%, (3) Mean: 51.9	Employed by large-scale company: 100%	(1) Stomach: 22%, lung: 13%, intestine: 11%, breast: 8%, female genitals: 5%, (2) NA, (3) NA	(1) Total: 80.7%, (2) Former company, (3) 365 days
Shionoya (2016) [20]	C, H	(1) 73, (2) Male: 42%, (3) 40–49: 8%, 50–59: 25%, 60–69: 44%, 70–79: 16%	Self-employed: 34%, regularly employed: 32%, part-time: 26%, temporary: 8%	(1) Breast: 27%, colon: 25%, female genitals: 12%, lung: 10%, liver/GB/pancreas: 10%, (2) NA, (3) Che at outpatient clinic: 100%	(1) Total: 63.0%, male: 81.8%, female: 47.5%, age 40–49: 50.0%, 50–59: 33.3%, 60–69: 71.9%, 70–79: 91.7%, self-employed: 88.0%, regularly employed: 73.9%, part-time: 15.8%, temporary: 66.7%, (2) NA, (3) NA
Nakamura (2017) [21]	C, H	(1) 213, (2) Female only, (3) Median: 45/48/48 (CC/EC/OC)	Part-time: 43%/42%/33%, regularly employed: 33%/34%/36%, self-employed: 16%/12%/21%, civil servant: 8%/12%/9% (CC/EC/OC)	(1) 53%/31%/15% (CC/EC/OC), (2) Early (I and II): 87%/90%/76%, advanced (III and IV): 13%/10%/24% (CC/EC/OC), (3) Ope: 38%/66%/21% (CC/EC/OC), Ope+Che/Rad: 36% (CC), Che/Rad: 26% (CC), Ope+Che: 34%/79% (EC/OC)	(1) 85.0%/85.0%/88.1%/78.8% (Total/CC/EC/OC), (2) Former worksite: 82%/85%/100% (CC/EC/OC), (3) 1 year or more following start of treatment
Takahara (2017) [22]	C, H	(1) 61, (2) ^‡^ Male: 36%, (3) ^‡^ (Male) 50–59: 25%, 60–69: 72%, (Female) 40–49: 35%, 50–59: 30%, 60–69: 30%	Regularly employed: 48%, non-regularly employed: 33%, self-employed: 20%	(1) ^‡^ Breast: 27%, colon: 24%, lung: 13%, lymphoma: 9%, female genitals: 8%, (2) ^‡^ I: 10%, II: 17%, III: 25%, IV: 31%, NA: 17%, (3) Che at outpatient clinic: 100%	(1) Total: 63.9%, regularly employed: 69.0%, non-regularly employed: 55.0%, self-employed: 66.7%, (2) Regularly employed: 51%, non-regularly employed: 28%, self-employed: 21%, (3) ^‡^ <2 years: 51%, 2–4 years: 28%, 4–6 years: 10%, 7+ years: 11%
Tomita (2017) [23]	C, H	(1) 84, (2) Female only, (3) Mean (SD): 55.4 (8.7), range: 31–77 (at time of survey)	Regularly employed: 67%, part-time: 33%	(1) Breast, (2) I: 36%, II: 45%, III: 13%, IV: 6%, (3) Ope: 98%, Che: 57%, HT: 74%, Rad: 67%	(1) 70.2%, (2) Regularly employed: 56%, part-time: 44% (3) Mean (SD): 62 (40) months, range: 10–201 months

Note: Figures are based on the data of subjects who were working at the time of cancer diagnosis. Nitta et al. (2015) [17] presented the subjects’ variables according to the cases. ^†^ Calculated on the assumption that homemakers were excluded. ^‡^ Data included those who were not working at the time of cancer diagnosis. ALND: axillary lymph node dissection; C: cross-sectional; CC: cervical cancer; Che: chemotherapy; CS: conserving surgery; EC: endometrial cancer; ET: endocrine therapy; GB: gallbladder; H: hospital-based study; HT: hormone therapy; LS: laparoscopy; LT: laparotomy; MTT: molecular targeted therapy; NA: not available; OC: ovarian cancer; Ope: operation; P: prospective; Rad: radiotherapy; RND: radical neck dissection; SD: standard deviation; TM: total mastectomy; W: worksite-based study.

**Table 2 ijerph-16-01461-t002:** Methodological quality assessment of selected studies.

First Author (Publication Year)	Risk of Bias Assessment Tool for Non-Randomized Studies (RoBANS) Domain and Risk of Bias					
	1. Selection of Participants (Selection Bias)	2. Confounding Variables (Selection Bias)	3. Measurement of Exposure (Cancer Diagnosis) (Performance Bias)	4. Blinding of Outcome (Return to Work) Assessment (Detection Bias)	5. Incomplete Outcome Data (Attrition Bias)	6. Selective Outcome Reporting (Reporting Bias)
Okumura (2006) [11]	High	High	Low	Low	High	Low
Shimada (2006) [12]	High	High	Low	Unclear	Unclear	High
Uchida (2009) [13]	High	High	Low	Low	High	High
Ozaki (2013) [14]	High	High	Low	Low	High	High
Suzuki (2014) [15]	High	High	Low	Low	Unclear	High
Ito (2015) [16]	High	High	Low	Low	High	Low
Nitta (2015) [17]	High	Low	Low	Low	Unclear	High
Matsuda (2015) [18]	High	Low	Low	Low	High	Low
Endo (2016) [19]	Low	High	Low	Low	Low	Low
Shionoya (2016) [20]	High	High	Low	Low	High	High
Nakamura (2017) [21]	High	Low	Low	Low	High	Low
Takahara (2017) [22]	High	Low	Low	Low	High	Low
Tomita (2017) [23]	High	Low	Low	Low	High	Low

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
