# Peer review of "Recent Status and Methodological Quality of Return-to-Work Rates of Cancer Patients Reported in Japan: A Systematic Review"

_ijerph, 2019, doi:10.3390/ijerph16081461_

Reviewer 1 Report

Main Contribution: The paper assesses the methodological quality of articles estimating return-to-work rates of cancer patients reported in Japan in the existing literature. The paper finds that in all 13 studies selected for review, there is a pervasive risk of selection bias, attrition bias, and reporting bias. Discussion: The authors argue convincingly on the source of bias for all the studies analyzed. In particular, the author make a strong case against the use of cross-sectional design. The “pars construens” however is not as well developed: the authors dedicate only one paragraph to generic suggestions on how to design studies that assess return-to-work rates of cancer patients. It would be beneficial to dedicate a specific section to the “ideal design” of studies assessing return-to-work rates of cancer patients. For example, the authors could review the literature on return-to-work rates for different pathologies in which return rates are estimated keeping into account the risk of selection, attrition, and reporting bias, or possibly investigate if there are studies on return-to-work rates for cancer patients outside of Japan that use a more suited design. To give an example, a related, older paper “Modified Work and Return to Work: A Review of the Literature” Journal of Occupational Rehabilitation, Vol. 8, No. 2, 1998, identifies higher quality studies and uses them to evaluate effectiveness of modified work programs. In a similar fashion, the authors should find higher quality studies and show how these studies deal with selection bias, attrition bias, and reporting bias. Note: sub-section 2.2 is completely disjoint from the rest of the paper. Calculating confidence intervals for each study’s return-to-work rates is a fine exercise, but what do we learn from it? The authors should elaborate more.

Author Response

Reply to Reviewer 1

>Main Contribution: The paper assesses the methodological quality of articles estimating return-to-work rates of cancer patients reported in Japan in the existing literature. The paper finds that in all 13 studies selected for review, there is a pervasive risk of selection bias, attrition bias, and reporting bias.

We appreciate you for reading our manuscript carefully.

>Discussion: The authors argue convincingly on the source of bias for all the studies analyzed. In particular, the author make a strong case against the use of cross-sectional design. The “pars construens” however is not as well developed: the authors dedicate only one paragraph to generic suggestions on how to design studies that assess return-to-work rates of cancer patients. It would be beneficial to dedicate a specific section to the “ideal design” of studies assessing return-to-work rates of cancer patients. For example, the authors could review the literature on return-to-work rates for different pathologies in which return rates are estimated keeping into account the risk of selection, attrition, and reporting bias, or possibly investigate if there are studies on return-to-work rates for cancer patients outside of Japan that use a more suited design. To give an example, a related, older paper “Modified Work and Return to Work: A Review of the Literature” Journal of Occupational Rehabilitation, Vol. 8, No. 2, 1998, identifies higher quality studies and uses them to evaluate effectiveness of modified work programs. In a similar fashion, the authors should find higher quality studies and show how these studies deal with selection bias, attrition bias, and reporting bias.

We proposed the ideal design of future studies in the last paragraph of the Discussion (page 12, lines 17 – 28 of our previous manuscript). Following your suggestion, we improved it to introduce the way of reducing the risk of selection, attrition, and reporting biases in more detail.

To reduce the risk of selection bias due to participant selection, a prospective cohort design is recommended. According to the review article you presented (added as reference no. 29), a prospective cohort design was employed to estimate the return-to-work rates of patients with injuries even in the 1980s and 90s. In fact, some research employed a prospective cohort design to estimate the rates of cancer patients. We have added this fact (Page 12, lines 18 – 24).

Regarding the selective reporting of the confounding variables and outcome, we introduced a recent paper that pointed out this issue in the hematology trials (reference no. 35). We have additionally introduced what researchers must do to deal with it (Page 12, lines 26 – 30).

To reduce the risk of attrition bias, Korean researchers used the national database to examine the outcome data, i.e., the employment status of cancer patients. We have introduced this (Page 12, lines 31 – 35).

Text revised:

(Page 12, lines 18 – 24) “Regarding selection bias due to participant selection, a cross-sectional design should be avoided, although it is also true in other countries that a cross-sectional design has often been employed in previous studies to estimate return-to-work rates of cancer patients [4–6]. Instead, prospective cohort studies must be conducted to determine all eligible participants at the time of cancer diagnosis. Some studies employed a prospective cohort design to examine the return to work of patients with injuries even in the 1980s and 90s [29] and that of cancer patients recently [30–34].”

(Page 12, lines 26 – 30) “Adequate trial registration and adherence to reporting guidelines would limit selective reporting of the confounding variables and outcome [35]. Researchers must register their trials prospectively, define the confounding variables and study outcomes explicitly, and address discrepancies between their own findings and existing ones honestly [35].”

(Page 12, lines 31 – 35) “A potential solution is a collaboration with the national database which involves all citizens. For example, researchers in Korea examined the employment status of cancer patients using the National Health Insurance administrative database [30,36,37]. This contributed to reducing the risk of attrition bias. No Japanese relevant studies have used such official data.”

References newly added:

29.           Krause, N.; Dasinger, L. K.; Neuhauser, F. Modified Work and Return to Work: A Review of the Literature. J. Occup. Rehabil. 1998, 8, 113–139.

30.           Choi, K.S.; Kim, E.J.; Lim, J.H.; Kim, S.G.; Lim, M.K.; Park, J.G.; Park, E.C. Job loss and reemployment after a cancer diagnosis in Koreans - a prospective cohort study. Psychooncology 2007, 16, 205–213.

31.           Johnsson, A.; Fornander, T.; Olsson, M.; Nystedt, M.; Johansson, H.; Rutqvist, L.E. Factors associated with return to work after breast cancer treatment. Acta Oncol. 2007, 46, 90–96.

32.           de Boer, A.G.; Verbeek, J.H.; Spelten, E.R.; Uitterhoeve, A.L.; Ansink, A.C.; de Reijke, T.M.; Kammeijer, M.; Sprangers, M.A.; van Dijk, F.J. Work ability and return-to-work in cancer patients. Br. J. Cancer 2008, 98, 1342–1347.

33.           Johnson, A.; Fornander, T.; Rutqvist, L.E.; Vaez, M.; Alexanderson, K.; Olsson, M. Predictors of return to work ten months after primary breast cancer surgery. Acta Oncol. 2009, 48, 93–98.

34.           Cooper, A.F.; Hankins, M.; Rixon, L.; Eaton, E.; Grunfeld, E.A. Distinct work-related, clinical and psychological factors predict return to work following treatment in four different cancer types. Psychooncology 2013, 22, 659–667.

35.           Wayant, C.; Scheckel, C.; Hicks, C.; Nissen, T.; Leduc, L.; Som, M.; Vassar, M. Evidence of selective reporting bias in hematology journals: A systematic review. PLoS One 2017, 12, e0178379.

36.           Park, J.H.; Park, E.C.; Park, J.H.; Kim, S.G.; Lee, S.Y. Job loss and re-employment of cancer patients in Korean employees: a nationwide retrospective cohort study. J. Clin. Oncol. 2008, 26, 1302–1309.

37.           Park J.H.; Park, J.H.; Kim, S.G. Effect of cancer diagnosis on patient employment status: a nationwide

>Note: sub-section 2.2 is completely disjoint from the rest of the paper. Calculating confidence intervals for each study’s return-to-work rates is a fine exercise, but what do we learn from it? The authors should elaborate more.

Point return-to-work rates estimated in studies with small sample sizes might not reflect the return-to-work rate of the corresponding population. We thus calculated the confidence intervals. We have added this point.

Text revised:

(Page 2, lines 25 – 28) “Some of the selected studies had small sample sizes. Point return-to-work rates estimated in such studies might not reflect the rate of the corresponding population. Therefore, we calculated a 95% confidence interval (CI) for each study’s return-to-work rate. We used the following formula, ..."

Reviewer 2 Report

Thank you for this submission. I truly enjoyed reading the paper, and was impressed with the rigour of the methodology described. One could argue, of course, that using further databases may had produced a more rounded perspective on the subject, but the submitted work is presented in a scientifically sound way. 

I wonder if the authors have considered adding a few thoughts about the limitations of the review regarding adding 'Japan' in the inclusion criteria for the English-written papers. Some studies may not include Japan in the title for example, yet the study is undertaken in Japan. 

Author Response

Reply to Reviewer 2

>Thank you for this submission. I truly enjoyed reading the paper, and was impressed with the rigour of the methodology described. One could argue, of course, that using further databases may had produced a more rounded perspective on the subject, but the submitted work is presented in a scientifically sound way.

We appreciate you for carefully reading and highly evaluating our manuscript.

>I wonder if the authors have considered adding a few thoughts about the limitations of the review regarding adding 'Japan' in the inclusion criteria for the English-written papers. Some studies may not include Japan in the title for example, yet the study is undertaken in Japan.

The term “Japan” must have been included in the abstract of the research conducted in Japan even if the title did not have the term. The PubMed, Scopus, CINAHL, and ICHUSHI web databases must have also detected such papers. A more possible issue is that the web databases did not contain the relevant papers which were written in English and published in little-known journals. We have added this issue.

Text revised:

(Page 12, lines 39 – 41) “We used major and reliable web databases to extract existing papers. However, we could possibly have failed to detect a few relevant papers which were written in English and published in little-known journals.”

Round  2

Reviewer 1 Report

My points have all been carefully addressed.